# Associations of air pollution exposures in preconception and pregnancy with birth outcomes and infant neurocognitive development: analysis of the Complex Lipids in Mothers and Babies (CLIMB) prospective cohort in Chongqing, China

Yingxin Chen [1], Tao Kuang,[2] Ting Zhang,[3] Samuel Cai,[4] John Colombo,[5] Alex Harper,[6] Ting-Li Han,[7,8] Yinyin Xia [9] John Gulliver,[10] Anna Hansell [1], Hua Zhang [11] Philip Baker[12]

For numbered affiliations see end of article.

**Correspondence to**
Dr Yinyin Xia;
100118@cqmu.edu.cn

## ABSTRACT

**Objectives** To investigate the associations of traffic-related air pollution exposures in early pregnancy with birth outcomes and infant neurocognitive development.

**Design** Cohort study.

**Setting** Eligible women attended six visits in the maternity clinics of two centres, the First Affiliated Hospital of Chongqing Medical University and Chongqing Health Centre for Women and Children.

**Participants** Women who were between 20 and 40 years of age and were at 11–14 weeks gestation with a singleton pregnancy were eligible for participation. Women were excluded if they had a history of premature delivery before 32 weeks of gestation, maternal milk allergy or aversion or severe lactose intolerance. 1273 pregnant women enrolled in 2015–2016 and 1174 live births were included in this analysis.

**Exposures** Air pollution concentrations at their home addresses, including particulate matter with diameter ≤2.5 µm ($PM_{2.5}$) and nitrogen dioxide ($NO_2$), during preconception and each trimester period were estimated using land-use regression models.

**Outcome measures** Birth outcomes (ie, birth weight, birth length, preterm birth, low birth weight, large for gestational age and small for gestational age (SGA) status) and neurodevelopment outcomes measured by the Chinese version of Bayley Scales of Infant Development.

**Results** An association between SGA and per-IQR increases in $NO_2$ was found in the first trimester (OR: 1.57, 95% CI: 1.06 to 2.32) and during the whole pregnancy (OR: 1.33, 99% CI: 1.01 to 1.75). Both $PM_{2.5}$ and $NO_2$ exposure in the 90 days prior to conception were associated with lower Psychomotor Development Index scores (β: −6.15, 95% CI: −8.84 to −3.46; β: −2.83, 95% CI: −4.27 to −1.39, respectively). Increased $NO_2$ exposure was associated with an increased risk of psychomotor development delay during different trimesters of pregnancy.

### STRENGTHS AND LIMITATIONS OF THIS STUDY

⇒ This study uniquely explored the impacts of both pre-conception and prenatal exposure to particulate matter with diameter ≤2.5 and nitrogen dioxide on neurodevelopmental outcomes in young infants, within an urban environment characterised by relatively high air pollution levels.

⇒ We developed a land use regression model to capture spatial and temporal variations of air pollution at individual level to reduce exposure misclassification.

⇒ Our sample size was relatively small, limiting the statistical power to assess several outcomes.

⇒ We defined exposure windows for clinically-defined trimesters; sensitive periods may be shorter or longer than 3 months, or may exist in the overlap of multiple trimesters.

**Conclusions** Increased exposures to $NO_2$ during pregnancy were associated with increased risks of SGA and psychomotor development delay, while increased exposures to both $PM_{2.5}$ and $NO_2$ pre-conception were associated with adverse psychomotor development outcomes at 12 months of age.

**Trial registration number** ChiCTR-IOR-16007700

## INTRODUCTION

Air pollution is a major environmental factor that has been linked to a range of adverse health outcomes in children. Maternal exposure to air pollutants during pregnancy, especially particulate matter (PM) with diameter ≤2.5 µm ($PM_{2.5}$) and nitrogen dioxide ($NO_2$), has been found to be associated with adverse birth outcomes,

including pre-term birth (PTB),[1] term low birth weight (TLBW),[2] and small for gestational age (SGA) status.[3] According to the developmental origins of health and disease hypothesis, prenatal exposures to air pollution may lead to adverse birth outcomes and subsequently increase the susceptibility to the development of certain diseases later in life.[4] A number of epidemiological studies have linked prenatal air pollution exposure with neurodevelopmental disorders such as autism spectrum disorder (ASD), attention deficit hyperactivity disorder and cognitive impairment.[5] Although the underlying biological mechanisms are still unclear, some studies indicated that prenatal air pollution exposure may induce systemic oxidative stress that triggers intra-uterine inflammation, leading to damage to several fetal organs, including the brain.[6 7]

It is also unclear whether the adverse effects of air pollution may start earlier before conception. Three months before conception was considered as a critical developmental window for gametogenesis. Air pollution exposure during the 3-month preconception period may have adverse effects on the gametogenesis of sperm[8 9] and ova cells.[10] Exposures to $PM_{2.5}$ in the preconception period have been associated with various neurodevelopmental outcomes, such as neural tube defects,[11] lower psychomotor development scores,[12] higher risk of ASD[13 14] and higher risk of intellectual disability.[15] Further research is required due to inconsistencies across studies in terms of studied health outcomes and exposure levels of air pollution.[12] Additionally, while there is growing evidence for the effects of preconception $PM_{2.5}$ exposure on the risk of adverse neurodevelopmental outcomes, no study to date has examined the effects of preconception $NO_2$ exposure. Exposure to $NO_2$ during pregnancy may be linked to compromised neural development in children, particularly affecting fine psychomotor skills.[16] Studying $PM_{2.5}$ along with $NO_2$ may allow us to explore how multiple pollutants affect birth outcomes and infant neurocognitive development independently and jointly. Moreover, both $PM_{2.5}$ and $NO_2$ are regulated traffic-related air pollutants in many countries. Understanding their impacts on birth and infant neurocognitive development can provide valuable insights for policymakers and public health authorities to develop effective air quality regulations and interventions.

Many studies have reported the effects of prenatal exposure to air pollution on neurodevelopmental function in children. However, the reported associations vary, due to the heterogeneous assessments of air pollution and neurodevelopmental outcomes.[5 17]

The current study leveraged the Complex Lipids in Mothers and Babies (CLIMB) cohort, a prospective birth cohort recruited in Chongqing, China,[18] with trimester-specific maternal $PM_{2.5}$ and $NO_2$ air pollution exposure derived from a spatio-temporal land use regression (LUR) model.[19] The aim of this analysis was to examine the associations between $PM_{2.5}$ and $NO_2$ exposures during pre-pregnancy and during pregnancy, with birth and infant neurocognitive development outcomes at 12 months of age.

A key aspect in all studies like this one is the accuracy of documenting exposure; a recent Chinese study determined air pollution exposure based on data from the nearest monitoring station[20] may not reflect the fine temporal and spatial variability of pollutant exposures among participants. Our study employed common air pollution exposure models based on advanced geographical information systems, to address some of the limitations of previous studies.[5]

In addition, the timing of exposure is also critical in determining the effects of exposure on developmental outcomes. Indeed, the evidence from previous studies on the sensitive time windows for exposure pre-pregnancy and during pregnancy remains inconclusive. Some studies have indicated that the early-to-mid pregnancy phase may be a critical period in terms of the impact of air pollution on neurodevelopment.[21 22] Early pregnancy is particularly important for neurogenesis and neuromigration, making it a susceptible period.[23] However, some studies reported stronger associations for middle or late pregnancy.[20 24 25] More studies identifying critical periods are needed to enhance our understanding of how pre-conception and prenatal air pollution exposure affect neurodevelopment. With this cohort, we are able to examine the effects of exposure pre-conception, at each trimester and the entire pregnancy.

## METHODS
### Study population
Participant recruitment in the CLIMB cohort has been described previously.[18] In brief, women who were between 20 and 40 years of age and were at 11–14 weeks gestation with a singleton pregnancy were eligible for participation. Women were excluded if they had a self-stated history of premature delivery before 32 weeks of gestation, maternal milk allergy or aversion or severe lactose intolerance.

From September 2015 to November 2016, a total of 1500 women were recruited into the cohort. Participants attended six visits at the First Affiliated Hospital of Chongqing Medical University and Chongqing Health Centre for Women and Children: 11–14 weeks' gestation (visit 1), 22–28 week's gestation (visit 2), 32–34 week's gestation (visit 3), at birth (visit 4), 6 weeks postnatal (visit 5) and 12 months postnatal (visit 6).

Women who withdrew from the study (n=146), terminated their pregnancy (n=29), miscarried (n=12) or were lost to follow-up (n=40) were excluded from the analysis, leaving a sample size of 1273 women. Analyses were restricted to mothers whose detailed residential addresses during pregnancy were known (figure 1). A total of 1174 live births were thus included in the pregnancy and neonatal outcomes analysis. Subsequently, at 1-year follow-up, 946 children were included in the analysis of neurodevelopment outcomes.

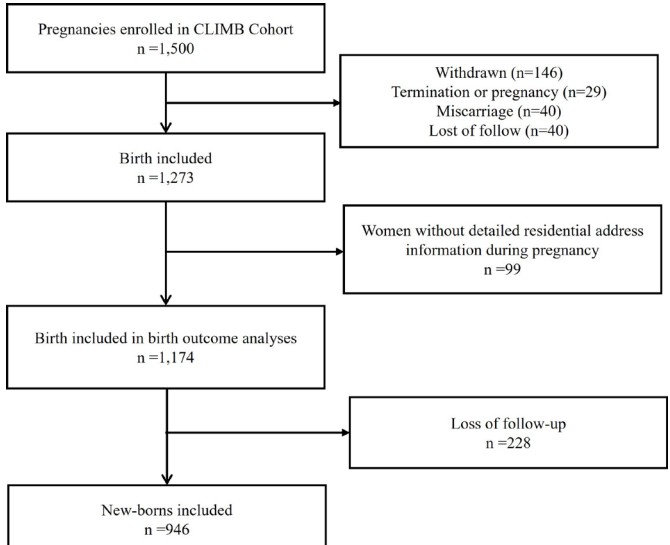

**Figure 1** Flow chart of the study population in CLIMB. CLIMB, Complex Lipids in Mothers and Babies.

## Study setting

The study area focused on the urban centre of the Chinese municipality of Chongqing (figure 2). The terrain of Chongqing is predominantly hilly and mountainous, with the core area located in a synclinal valley at the confluence of the Yangtze River and the Jialing River.[26] The urban core of Chongqing, our study area, has a population of approximately 6.52 million people, a land area of 5472 square kilometres and 4.62 million vehicles.[27] It shows a higher population density of approximately 1191 people per square kilometre and a lower number of motor vehicles of 0.71 per capita. The urban core of Chongqing used to have multiple old industries with higher $NO_2$ and $PM_{2.5}$ emissions, including the Chongqing Iron and Steel Company in Dadukou district and the Chongqing Thermal Power Plant in Jiulongpo district, both of which have been relocated to rural areas in Chongqing. The main sources of pollution in the area now include traffic-related emissions, construction activities and anthropogenic sources such as outdoor grilling and emissions from food establishments.[28] The coverage rate of the urban population with access to gas in Chongqing was 95.34%,[27] suggesting a low reliance on biomass cookstoves in urban areas.

## Exposure assessment

The address of participants was collected at the first visit. Exposure assessment based on spatiotemporal LUR models for $PM_{2.5}$ and $NO_2$ were developed for the study region. A description of the methodology of exposure modelling has been reported previously.[19] Briefly, the models included both spatial and temporal components of exposure. $PM_{2.5}$ and $NO_2$ concentration data were collected from 17 routine monitoring sites operated by the Chongqing Environmental Monitoring Center in

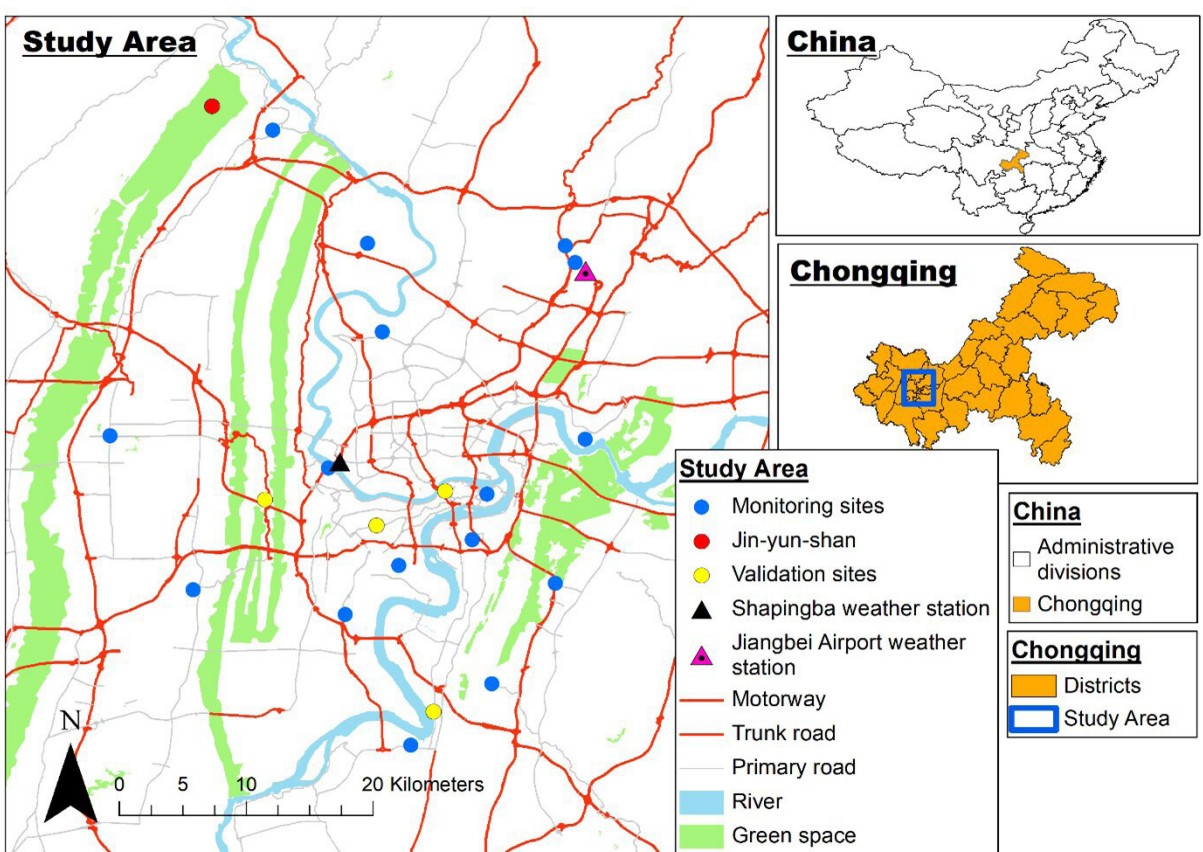

**Figure 2** Study area and location of monitoring sites (OpenStreetMap contributors, 2015; https://data.nextgis.com/en/region/CN-50/).

2015–2016. For the spatial component of models, we calculated the annual average concentrations of each pollutant in 2015 and fit linear regression models using five groups of geographical data (road network, land use, topography, vegetation and population density) as spatial predictor variables. For the temporal component of models, we calculated the residuals from the spatial component at each monitoring site on a daily basis by subtracting the predicted annual average concentration from the observed daily average concentrations measured in 2015 and 2016, and then fitted generalised additive models (GAM) using seven groups of meteorological data (temperature, amount of rainfall, rainfall events, relative humidity, horizontal visibility, wind direction and wind speed) as temporal predictor variables. The meteorological variables were used to account for the influence of weather on the change in air pollution concentration over time. To account for the remaining spatial autocorrelation, the smoothed terms of longitude and latitude were fit to spatiotemporal residuals which were calculated by subtracting the sum of the spatial temporal predictions from the measured daily average concentrations in 2015 and 2016. The performance of the $PM_{2.5}$ spatiotemporal models was good (correlation (COR)-$R^2$: 0.72) and the $NO_2$ spatiotemporal model was low (COR-$R^2$: 0.39) when providing concentration estimates in absolute terms.

Combining the family address coordinates of each pregnant woman and the gestation period of the pregnancy (calculated from the date of the last menstrual period to the date of delivery), we used this spatiotemporal model to estimate the average exposure of each pregnant woman in 90 days prior to pregnancy (90D), first trimester (T1), second trimester (T2), third trimester (T3) and whole pregnancy period (WP), respectively.

## Outcomes
### Birth outcomes
Birth outcomes were determined by experienced obstetricians and abstracted from the medical records. Birth outcomes included: birth weight (in grams), birth length (in centimetres), PTB, low birth weight (LBW), large for gestational age (LGA) and SGA status.[29] PTB was defined as delivery before 37 weeks. LBW was defined as weighing less than 2500 g at birth. LGA and SGA were indicated by birth weight greater than and less than the 90th and 10th percentile within this study for the gestational age by sex, respectively.[30] TLBW was not considered due to a small sample size of only eight cases.

### Neurodevelopment outcomes
The Chinese version of Bayley Scales of Infant Development (CBSID) was used to assess mental and psychomotor development for infants in this study. The CBSID is appropriate for the evaluation of infants from 2 to 30 months old[31] and takes into consideration each infant's age in days. Infants were assessed at around 12 months (range from 11 months and 15 days to 12 months and 15 days) by a trained examiner, with ages corrected for

preterm birth. These scales have been formally adapted to the Chinese language and locally standardised to become culturally appropriate, with two main indexes: the Mental Development Index (MDI) and the Psychomotor Development Index (PDI). The MDI component comprised 163 items and assessed age-appropriate items related to cognitive functioning, personal and social development and language development (see online supplemental etable 1). The PDI component comprised 81 items and assessed age-appropriate fine and gross motor skills (see online supplemental etable 2). The test provided raw scores for mental and psychomotor development that were converted to standardised (in terms of age in days) MDI and PDI scores, based on norms for the Chinese population. As with other forms of the Bayley test these index scores have a mean of 100, and an SD of 15, with a lower score reflecting poorer performance.[32] If an infant refused to cooperate with the examiners to finish the task, a second assessment was arranged within 2 weeks. If the infant could not cooperate at the second BSID assessment, their data were classified as missing. In addition to the continuous scores, we define mental developmental delay (MDD) and psychomotor developmental delay (PDD) if the score is less than 85.[33]

## Covariates
Socio-demographic data were collected through interviews by trained nurses. The following potential confounders were identified: maternal age at enrolment (in years), infant sex (male/female), maternal body mass index (BMI) at 11–14 weeks' gestation ($kg/m^2$), parity (yes/no), monthly household income level (categorised as: <¥2000, ¥2000 to ¥7000, ¥7000 to ¥10 000 or >¥10 000), season of birth (categorised as: Spring (March to May), Summer (June to August), Autumn (September to November) or Winter (December to February)). Season of birth was taken into consideration because air pollution and related environmental factors, such as temperature and humidity, may vary across different seasons (ie, air pollution levels tend to be higher during winter). Some studies suggest that the season of birth may indirectly influence cognitive function through factors such as seasonal differences in food availability affecting maternal nutrition during pregnancy, sunlight exposure impacting maternal vitamin D levels and children's early-life indoor and outdoor activities. Marital status (single/married) and smoking or drinking during pregnancy (yes/no) were not taken into account in this analysis because of the homogeneity of the study population (ie, 98.6% of women were married and 99.6% of women reported not smoking or drinking alcohol during pregnancy). We did not adjust dietary supplements during pregnancy because all pregnant women routinely take folic acid in this cohort.

## Statistical analyses
Data were described in terms of mean±SD or median (IQR) for continuous variables, or as percentages for

categorical variables. Modelled $PM_{2.5}$ and $NO_2$ exposure levels in 90D, T1, T2, T3 and WP were considered separately. We examined the Spearman correlation between each of the exposures in the different pregnancy periods. For birth outcomes, multivariable linear regression was used for continuous outcomes (eg, birth weight and birth length) to estimate β coefficient and their 95% CIs and multivariable logistic regression for binary outcomes (eg, PTB, LBW, LGA and SGA status) to estimate OR and 95% CIs. For mental and psychomotor development (eg, MDI and PDI scores), multivariable linear regression models were fit to estimate the β coefficient and their 95% CIs. We also conducted multivariable logistic regression analysis for binary neurodevelopment outcomes (ie, MDD and PDD). Models were adjusted for maternal age at enrolment, infant sex, maternal BMI at 11–14 weeks gestation, primiparity, monthly household income level and season of birth. We also ran co-exposure models to estimate associations of one air pollutant while additionally adjusting for the other air pollutant (ie, $PM_{2.5}$ effects in T1 adjusted for $NO_2$ in T1). Effect estimates are reported for each IQR increase of $PM_{2.5}$ and $NO_2$. All analyses were performed using Stata V.17. A p value of <0.05 was considered statistically significant to address multiple comparisons in the analyses.

### Patient and public involvement
None.

## RESULTS
### Study participants
Participant characteristics are presented in table 1. Of those participating women, the mean age was 28.7 years and the mean BMI was $21.5 \text{kg/m}^2$. 98.0% of women were of Han ethnicity, 77.9% were primiparous and 67.6% had completed tertiary education. 33 (2.8%), 30 (2.6%), 108 (9.2%), 84 (7.2%) of the 1174 births considered in this analysis were classified as PTB, LBW, LGA and SGA, respectively. For those 946 children who completed the BSID test, the mean MDI score was 94.7 (SD: 17.7) and the mean PDI score was 87.4 (SD: 14.9). The proportions of participants with MDD (MDI<85) and PDD (PDI<85) were 27.1% and 42.4%, respectively.

### Exposure assessment
Median $PM_{2.5}$ exposure concentrations were $57.31 \mu\text{g/m}^3$ (IQR: 5.76) and median $NO_2$ exposure levels were $50.46 \mu\text{g/m}^3$ (IQR: 5.51) during the whole pregnancy period (online supplemental etable 3). For $PM_{2.5}$, the concentration in the pre-conception and T1 were considerably lower than other periods, close to $10 \mu\text{g/m}^3$. The between-trimester and 90D values for $NO_2$ were generally moderately correlated (Pearson's r>0.5). The correlation coefficients of $PM_{2.5}$ were more variable between time periods reflecting the high variability of $PM_{2.5}$ concentrations, with values ranging from −0.78 to +0.68. Correlations between $PM_{2.5}$ and $NO_2$ in the same pregnancy period were moderately correlated (Pearson's r~0.6, online supplemental etable 4).

### Association with birth outcomes
In the unadjusted models (online supplemental etable 5), higher exposure concentrations of $PM_{2.5}$ in T3 were significantly associated with lower birth length (β: −0.32, 95% CI: −0.51 to −0.13; per IQR increase). We also observed that increased $NO_2$ in T3 was significantly associated with lower birth length (β: −0.16, 95% CI: −0.32 to −0.01; per IQR). A risk between SGA and increases in $NO_2$ (per IQR) was found in T2 (OR: 1.46, 95% CI: 1.10 to 1.93), T3 (OR: 1.58, 95% CI: 1.14 to 2.18) and in the whole pregnancy period (OR: 1.44, 95% CI: 1.13 to 1.85). We observed no evidence of associations of $NO_2$ with overall birth weight, birth length and other adverse birth outcomes (eg, PTB, LBW and LGA).

In the adjusted models (table 2), we found increased effect size for $NO_2$ and SGA in T2 (OR: 1.57, 95% CI: 1.06 to 2.32), and slightly reduced effects size for $NO_2$ and SGA in the whole pregnancy period (OR: 1.33, 95% CI: 1.01 to 1.75) compared with the unadjusted model. We observed no evidence of associations with birth length in the adjusted models. After co-adjustment for $PM_{2.5}$ (see online supplemental etable 6), the association of $NO_2$ with SGA was also found in T1 (OR: 1.70, 95% CI: 1.07 to 2.69), T3 (OR: 1.77, 95% CI: 1.08 to 2.91) and in the whole pregnancy period (OR: 1.60, 95% CI: 1.15 to 2.23).

### Association with infant neurodevelopment outcomes
In unadjusted models, $PM_{2.5}$ exposure in the 90 days prior to conception was associated with lower MDI and PDI scores in offspring (β: −3.54, 95% CI: −5.37 to −1.71; β: −3.42, 95% CI: −4.96 to −1.89) (table 3). We also observed an unexpected positive association between $PM_{2.5}$ exposures in the second trimester with MDI (β: 4.21, 95% CI: 2.43 to 6.00) and PDI (β: 2.63, 95% CI: 1.12 to 4.14). Exposure to $NO_2$ was associated with lower MDI (−1.90, 95% CI: −3.36 to −0.44) and PDI in the 90 days prior to conception (−2.86, 95% CI: −4.08 to −1.65). $NO_2$ exposure was also associated with lower PDI scores in T3 (−1.97, 95% CI: −3.29 to −0.65) and in the whole pregnancy periods (−1.08, 95% CI: −2.11 to −0.05). We did not observe any association between $NO_2$ and MDI in any pregnancy periods.

In the adjusted models (table 3), we found that $PM_{2.5}$ exposure in the 90 days prior to conception was associated with lower PDI scores (β: −6.15, 95% CI: −8.84 to −3.46). Similarly, there was also a significant association between increased $NO_2$ exposure and lower PDI score in the 90 days prior to conception (β: −2.83, 95% CI: −4.27 to −1.39), T1 (β: −1.91, 95% CI: −3.37 to −0.46), T3 (β: −1.92, 95% CI: −3.57 to −0.26) and whole pregnancy period (β: −1.15, 95% CI: −2.19 to −0.11). The positive association between $PM_{2.5}$ exposures in the second trimester with PDI (β: 3.76, 95% CI: 1.27 to 6.24) remained. We did not observe any association with MDI in any pregnancy periods.

**Table 1** Characteristics of the study sample in the Complex Lipids in Mothers and Babies cohort (N=1174)

| Characteristic of mother | N | n (%)/ mean±SD | Characteristic of child | N | n (%) /mean±SD |
|---|---|---|---|---|---|
| Maternal age (years) | 1174 | 28.7±3.5 | Gestational week (week) | 1174 | 39.4±1.5 |
| BMI (kg/m$^2$) | 1174 | 21.5±2.9 | Birth weight (g) | 1165 | 3314.4±428.8 |
| Han ethnicity (%) | 1174 | | Birth length (cm) | 1149 | 49.7±1.9 |
| Yes | | 1151 (98.0) | Newborn sex | 1172 | |
| No | | 23 (2.0) | Female | | 561 (47.9) |
| Marital status (%) | 1174 | | Male | | 611 (52.1) |
| Single | | 16 (1.4) | Birth outcomes | | |
| Married | | 1158 (98.6) | Preterm birth | 1174 | |
| Primiparity (%) | 1174 | | Yes | | 33 (2.8) |
| Yes | | 914 (77.9) | No | | 1141 (97.2) |
| No | | 260 (22.1) | Low birth weight | 1174 | |
| History of miscarriage or abortion (%) | 1174 | | Yes | | 30 (2.6) |
| Yes | | 553 (47.1) | No | | 1141 (97.2) |
| No | | 621 (52.9) | Large for gestational age | 1174 | |
| Smoking/drinking during pregnancy (%) | 1174 | | Yes | | 108 (9.2) |
| Yes | | 5 (0.4) | No | | 1066 (90.8) |
| No | | 1169 (99.6) | Small for gestational age | 1174 | |
| Education level | 946 | | Yes | | 84 (7.2) |
| Low: high school or below | | 306 (32.3) | No | | 1090 (92.8) |
| High: college/uni or above | | 640 (67.6) | BSID test | 946 | |
| Job | 946 | | MDI score | | 94.7±17.7 |
| Full-time | | 762 (80.5) | PDI score | | 87.4±14.9 |
| Housewife | | 82 (8.7) | Mental development | 946 | |
| Others | | 102 (10.8) | Delay (MDI<85) | | 276 (27.1) |
| Household income (monthly) | 946 | | Normal (MDI≥85) | | 741 (72.9) |
| <¥2000 | | 186 (19.7) | Psychomotor development | 946 | |
| ¥2000 to ¥4000 | | 329 (34.8) | Delay (PDI<85) | | 431 (42.4) |
| ¥4000 to ¥7000 | | 292 (30.9) | Normal (PDI≥85) | | 586 (57.6) |
| ¥7000 to ¥10 000 | | 139 (14.7) | Season of birth | 1174 | |
| | | | Spring (March to May) | | 411 (35.01) |
| | | | Summer (June to August) | | 263 (22.40) |
| | | | Autumn (September to November) | | 198 (16.87) |
| | | | Winter (December to February) | | 302 (25.72) |

BMI, body mass index; BSID, Bayley Scales of Infant Development ; MDI, Mental Development Index ; PDI, Psychomotor Development Index
.

In the co-exposure models (table 3), $PM_{2.5}$ exposure in the 90 days prior to conception was associated with lower PDI scores (β: −4.74, 95% CI: −7.73 to −1.75). We also observed a positive association between $PM_{2.5}$ exposures in the second trimester with PDI (β: 5.51, 95% CI:2.73 to 8.28). Exposure to $NO_2$ was significantly associated with lower PDI in 90D (β: −1.72, 95% CI: −3.31 to −0.12), T1 (β: −1.80, 95% CI: −3.46 to −0.15), T2 (β: −2.11, 95% CI: −3.63 to −0.60), T3 (β: −1.92, 95% CI: −3.76 to −0.09)

and whole pregnancy period (β: −1.68, 95% CI: −2.89 to −0.46).

In the adjusted model, the risk of PDD was found to increase by 112% and 42% with each per-IQR increase in $PM_{2.5}$ (OR: 2.12, 95% CI: 1.45 to 3.11) and $NO_2$ (OR: 1.42, 95% CI: 1.16 to 1.75) in the 90 days prior to conception (table 4). There was also a significant association between increased $NO_2$ exposure and the risk of PDD in T1 (OR: 1.29, 95% CI: 1.05 to 1.58), T3 (OR: 1.27 to 95% CI: 1.01

**Table 2** Associations between PM$_{2.5}$ and NO$_2$ exposure in different pregnancy periods and adverse birth outcomes (adjusted models)

| Per IQR increase in | | Mean difference | | ORs | | | |
|---|---|---|---|---|---|---|---|
| | | Birth weight, g (95% CI) | Birth length, cm (95% CI) | PTB (case: 33) (95% CI) | LBW (case: 30) (95% CI) | LGA (case: 108) (95% CI) | SGA (case: 84) (95% CI) |
| | | (N=941) | (N=927) | (N=945) | (N=945) | (N=945) | (N=945) |
| Estimated exposure to PM$_{2.5}$ | 90 days prior to conception | 59.73 (−16.52 to 135.98) | 0.15 (−0.176 to 0.48) | 0.24 (0.06 to 1.00) | 0.49 (0.18 to 1.29) | 1.40 (0.72 to 2.71) | 1.66 (0.75 to 3.68) |
| | First trimester | 6.21 (−73.79 to 86.20) | 0.04 (−0.308 to 0.388) | 0.88 (0.28 to 2.80) | 0.76 (0.21 to 2.81) | 0.86 (0.45 to 1.67) | 1.33 (0.58 to 3.04) |
| | Second trimester | −37.64 (−107.73 to 32.44) | 0.02 (−0.283 to 0.326) | 1.62 (0.53 to 4.96) | 1.34 (0.38 to 4.68) | 1.00 (0.55 to 1.83) | 0.94 (0.50 to 1.76) |
| | Third trimester | 4.20 (−73.17 to 81.57) | −0.17 (−0.509 to 0.162) | 0.92 (0.29 to 2.90) | 0.92 (0.30 to 2.85) | 1.29 (0.65 to 2.53) | 0.83 (0.42 to 1.66) |
| | Total pregnancy | 8.01 (−41.10 to 57.11) | 0.02 (−0.198 to 0.230) | 0.77 (0.38 to 1.54) | 0.62 (0.31 to 1.25) | 1.15 (0.75 to 1.77) | 0.84 (0.52 to 1.35) |
| Estimated exposure to NO$_2$ | 90 days prior to conception | −1.03 (−41.88 to 39.81) | −0.04 (−0.215 to 0.139) | 0.84 (0.45 to 1.57) | 1.04 (0.54 to 1.98) | 1.31 (0.91 to 1.88) | 1.45 (0.99 to 2.12) |
| | First trimester | −9.78 (−50.84 to 31.28) | 0.04 (−0.133 to 0.223) | 0.90 (0.49 to 1.65) | 1.03 (0.56 to 1.91) | 1.21 (0.85 to 1.72) | **1.57 (1.06 to 2.32)** |
| | Second trimester | −20.82 (−59.11 to 17.47) | −0.06 (−0.222 to 0.112) | 1.31 (0.73 to 2.34) | 1.34 (0.75 to 2.40) | 1.21 (0.86 to 1.70) | 1.36 (0.95 to 1.95) |
| | Third trimester | −9.50 (−56.00 to 36.99) | −0.01 (−0.213 to 0.191) | 0.79 (0.40 to 1.59) | 0.95 (0.47 to 1.94) | 1.42 (0.94 to 2.13) | 1.51 (0.97 to 2.36) |
| | Total pregnancy | −8.45 (−37.73 to 20.83) | 0.00 (−0.125 to 0.130) | 0.97 (0.62 to 1.51) | 1.04 (0.66 to 1.64) | 1.20 (0.93 to 1.56) | **1.33 (1.01 to 1.75)** |

All significant findings in the table are bold.

Models adjusted for maternal age at enrolment, infant's sex, maternal body mass index at 11–14 weeks' gestation, primiparity, monthly household income level and season of births.

LBW, low birth weight; LGA, large for gestational age; NO$_2$, nitrogen dioxide ; PM$_{2.5}$, particulate matter with diameter ≤2.5; PTB, pre-term birth ; SGA, small for gestational age.

**Table 3** Associations between PM$_{2.5}$ and NO$_2$ exposure in different pregnancy periods and continuous Bayley Scales of Infant Development scores

| | | Crude models | | Adjusted models* | | Co-exposure models† | |
|---|---|---|---|---|---|---|---|
| **Per IQR increase in** | | MDI (95% CI) (N=946) | PDI 95% CI (N=946) | MDI (95% CI) (N=945) | PDI 95% CI (N=945) | MDI (95% CI) (N=945) | PDI 95% CI (N=945) |
| Estimated exposure to PM$_{2.5}$ | 90 days prior to conception | **−3.54 (−5.37 to 1.71)** | **−3.42 (−4.96 to 1.89)** | −1.98 (−5.19 to 1.23) | **−6.15 (−8.84 to 3.46)** | −1.73 (−5.30 to 1.85) | **−4.74 (−7.73 to 1.75)** |
| | First trimester | −1.07 (−2.93 to 0.79) | 0.04 (−1.52 to 1.61) | −1.66 (−5.02 to 1.70) | −2.11 (−4.95 to 0.73) | −2.84 (−6.65 to 0.97) | −0.45 (−3.67 to 2.76) |
| | Second trimester | **4.21 (2.43 to 6.00)** | **2.63 (1.12 to 4.14)** | 3.79 (0.85 to 6.73) | **3.76 (1.27 to 6.24)** | 4.19 (0.89 to 7.49) | **5.51 (2.73 to 8.28)** |
| | Third trimester | −1.43 (−3.41 to 0.55) | **−1.76 (−3.42 to 0.10)** | −2.73 (−5.99 to 0.53) | −1.37 (−4.12 to 1.39) | −3.84 (−7.46 to 0.22) | 0.04 (−3.02 to 3.09) |
| | Total pregnancy | **1.64 (0.06 to 3.21)** | 0.5 (−0.82 to 1.83) | −0.27 (−2.34 to 1.80) | 0.23 (−1.52 to 1.98) | −0.85 (−3.28 to 1.57) | 1.69 (−0.35 to 3.73) |
| Estimated exposure to NO$_2$ | 90 days prior to conception | **−1.90 (−3.36 to 0.44)** | **−2.86 (−4.08 to 1.65)** | −0.72 (−2.43 to 1.00) | **−2.83 (−4.27 to 1.39)** | −0.31 (−2.22 to 1.60) | **−1.72 (−3.31 to 0.12)** |
| | First trimester | −0.08 (−1.57 to 1.42) | −1.17 (−2.43 to 0.08) | 0.59 (−1.14 to 2.32) | **−1.91 (−3.37 to 0.46)** | 1.28 (−0.68 to 3.24) | **−1.80 (−3.46 to 0.15)** |
| | Second trimester | **1.81 (0.41 to 3.22)** | 0.00 (−1.18 to 1.18) | 0.56 (−1.05 to 2.17) | −0.75 (−2.11 to 0.61) | −0.48 (−2.28 to 1.33) | **−2.11 (−3.63 to 0.60)** |
| | Third trimester | 0.04 (−1.54 to 1.62) | **−1.97 (−3.29 to 0.65)** | 0.51 (−1.45 to 2.47) | **−1.92 (−3.57 to 0.26)** | 1.52 (−0.66 to 3.69) | **−1.92 (−3.76 to 0.09)** |
| | Total pregnancy | 0.67 (−0.56 to 1.89) | **−1.08 (−2.11 to 0.05)** | 0.41 (−0.83 to 1.64) | **−1.15 (−2.19 to 0.11)** | 0.67 (−0.77 to 2.12) | **−1.68 (−2.89 to 0.46)** |

All significant findings in the table are bold.
*Models adjusted for maternal age at enrolment, infant's sex, maternal BMI at 11–14 weeks' gestation, primiparity, monthly household income level and season of births.
†Models adjusted for maternal age at enrolment, infant's sex, maternal BMI at 11–14 weeks' gestation, primiparity, monthly household income level and season of births, and adjusted for the other air pollutant.
BMI, body mass index; MDI, Mental Development Index ; NO$_2$, nitrogen dioxide ; PDI, Psychomotor Development Index; PM$_{2.5}$, particulate matter with diameter ≤2.5.

**Table 4** Associations between $PM_{2.5}$ and $NO_2$ exposure in different pregnancy periods and mental and psychomotor developmental delay

| Per IQR increase in | | Crude models | | Adjusted models* | | Co-exposure models† | |
|---|---|---|---|---|---|---|---|
| | | MDD (95% CI) (N=946) | PDD (95% CI) (N=946) | MDD (95% CI) (N=945) | PDD (95% CI) (N=945) | MDD (95% CI) (N=945) | PDD (95% CI) (N=945) |
| Estimated exposure to $PM_{2.5}$ | 90 days prior to conception | **1.45 (1.16 to 1.83)** | **1.49 (1.20 to 1.83)** | 0.95 (0.64 to 1.42) | **2.12 (1.45 to 3.11)** | 0.97 (0.63 to 1.51) | **1.78 (1.17 to 2.71)** |
| | First trimester | 1.05 (0.83 to 1.33) | 1.04 (0.84 to 1.28) | 1.14 (0.73 to 1.79) | 1.42 (0.96 to 2.11) | 1.35 (0.80 to 2.25) | 1.16 (0.74 to 1.82) |
| | Second trimester | **0.63 (0.49 to 0.80)** | **0.77 (0.63 to 0.95)** | 0.81 (0.54 to 1.22) | 0.72 (0.51 to 1.02) | 0.83 (0.52 to 1.31) | **0.57 (0.38 to 0.85)** |
| | Third trimester | 1.23 (0.96 to 1.58) | 1.19 (0.95 to 1.49) | 1.25 (0.82 to 1.90) | 1.17 (0.80 to 1.70) | 1.39 (0.87 to 2.23) | 0.98 (0.64 to 1.49) |
| | Total pregnancy | 0.84 (0.69 to 1.03) | 0.98 (0.82 to 1.18) | 1.07 (0.82 to 1.39) | 1.07 (0.84 to 1.35) | 1.17 (0.86 to 1.59) | 0.90 (0.68 to 1.20) |
| Estimated exposure to $NO_2$ | 90 days prior to conception | 1.20 (0.99 to 1.45) | **1.41 (1.19 to 1.67)** | 0.97 (0.77 to 1.21) | **1.42 (1.16 to 1.75)** | 0.97 (0.76 to 1.25) | 1.24 (0.99 to 1.56) |
| | First trimester | 0.97 (0.80 to 1.17) | 1.18 (0.99 to 1.40) | 0.91 (0.72 to 1.13) | **1.29 (1.05 to 1.58)** | 0.84 (0.65 to 1.09) | 1.24 (0.99 to 1.57) |
| | Second trimester | **0.79 (0.66 to 0.95)** | 1.04 (0.88 to 1.22) | 0.94 (0.76 to 1.15) | 1.14 (0.95 to 1.38) | 0.98 (0.77 to 1.24) | **1.31 (1.06 to 1.63)** |
| | Third trimester | 1.04 (0.85 to 1.28) | **1.25 (1.04 to 1.50)** | 0.94 (0.73 to 1.21) | **1.27 (1.01 to 1.60)** | 0.86 (0.65 to 1.14) | 1.28 (0.99 to 1.65) |
| | Total pregnancy | 0.92 (0.79 to 1.07) | **1.16 (1.01 to 1.33)** | 0.94 (0.80 to 1.11) | **1.17 (1.02 to 1.36)** | 0.9 (0.75 to 1.08) | **1.21 (1.02 to 1.43)** |

All significant findings in the table are bold.
*Models adjusted for maternal age at enrolment, infant's sex, maternal BMI at 11–14 weeks' gestation, primiparity, monthly household income level and season of births.
†Models adjusted for maternal age at enrolment, infant's sex, maternal BMI at 11–14 weeks' gestation, primiparity, monthly household income level and season of births, and adjusted for the other air pollutant.
BMI, body mass index; MDD, mental developmental delay; $NO_2$, nitrogen dioxide ; PDD, psychomotor developmental delay; $PM_{2.5}$, particulate matter with diameter ≤2.5.

to 1.60) and the whole pregnancy period (OR: 1.17, 95% CI: 1.02 to 1.36). We did not observe any association with MDD in any pregnancy periods.

## DISCUSSION

We analysed associations between modelled $PM_{2.5}$ and $NO_2$ pre-pregnancy and during pregnancy with birth and neurodevelopment outcomes in singleton children born in a southwestern metropolis of China in 2015–2016. We found the likelihood of SGA increased by 33% per IQR higher exposure to $NO_2$ in the whole pregnancy periods after adjusted for maternal age at enrolment, infant's sex, maternal BMI at 11–14 weeks' gestation, primiparity, monthly household income level and season of births and $PM_{2.5}$. For childhood cognitive development, increased exposure to $PM_{2.5}$ and $NO_2$ in the 90 days prior to conception were both associated with lower PDI scores, with the effect size per IQR being higher for $PM_{2.5}$ than for $NO_2$. Increased $NO_2$ exposure was associated with an increased risk of PDD during different trimesters of pregnancy.

Many studies from other geographical areas, including Europe,[34–36] the USA[21 25] and Asia,[22 37–39] have found significant associations between prenatal air pollution exposure and a variety of adverse neurodevelopmental outcomes. Our finding of a negative association between prenatal $NO_2$ air pollution exposure and infant neurocognitive development is consistent with these reports. A recent Chinese birth cohort study of 15 778 child–mother pairs in Foshan reported that maternal $NO_2$ exposure during pregnancy was associated with an increased risk of suspected developmental delay (OR: 1.06, 95% CI: 0.94 to 1.19) measured by a five-domain scale and developmental quotient.[22] A birth cohort study of 520 mother–child pairs in South Korea reported that maternal $NO_2$ exposure during pregnancy was associated with impairment of psychomotor development (β=−1.30, p=0.05) but—as in the present study—not with cognitive function (β=−0.84, p=0.20).[37] However, results from previous research varied by air pollutants. For example, a Chinese study of 1193 mother–newborn pairs in Changsha found significant associations between $PM_{2.5}$ exposure in trimester two and lower neurobehavioural developmental scores, while other air pollutants such as $PM_{10}$, carbon monoxide and sulphur dioxide had null or even reverse associations. In this study, we observed that the negative effect of $NO_2$ exposure during pregnancy on PDI is significant at a 5% level; this negative effect of $NO_2$ still remained after adjustment for $PM_{2.5}$. This heterogeneity may relate to the temporality of exposure assessment, types of outcome assessment instruments or evaluators and the levels of air pollution. In addition, air pollution mixtures may have differed among the study regions, thus there are several potential explanations for the heterogeneity of the findings. We also observed negative correlations between certain exposures, indicating the need to consider potential collinearity in our two-pollutant models. In Chongqing, a major industrial city in southwest China, air

pollution may come from industrial and traffic emissions, construction activities and dust and negative correlations may occur if different sources contribute disproportionately to each pollutant. Their correlations may also be affected by seasonal changes and variations in weather patterns. Future research should also explore the impact of source-specific air pollution on children's cognitive health.

To date, most studies on prenatal air pollution exposure and child neurodevelopment have been conducted in developed countries with relatively low levels of air pollution. In this study, the level of air pollution was higher (median $PM_{2.5}$: 57.31 μg/m³, IQR: 5.76; median $NO_2$: 50.46 μg/m³, IQR: 5.51) compared with studies in developed countries such as Europe and the USA. In a multicentre European cohort, the mean $PM_{2.5}$ and $NO_2$ exposure concentration during pregnancy were 13.4 μg/m³ and 11.5 μg/m³.[34] Researchers found that the psychomotor development score significantly decreased by 0.68 points (95% CI: −1.25 to −0.11) for every 10 μg/m³ increase in $NO_2$, and there was also a non-significant decrease of 1.64 points (95% CI: −3.47 to 0.18) for every 5 μg/m³ increase in $PM_{2.5}$ during pregnancy.[34] Factors such as the types of pollutants and concentrations may differ between China and other regions with a lower air pollution level, leading to variations in the observed effects.

Contrary to expectations, we found significant positive associations between prenatal exposure to $PM_{2.5}$ air pollution in the second trimester and PDI. However, no association was observed between $PM_{2.5}$ exposures in the second trimester and the risk of PDD. Given the existing literature and the conflicted observation here, we believe that this is likely to be spurious/sample specific. Some plausible explanations include the uneven distribution of PDI scores, the potentially inappropriate selection of the cut-off value of 85 (which may not effectively discriminate between groups), or the possibility that the observed outcome occurred by chance. Several epidemiological studies have reported associations between prenatal exposure to high levels of $PM_{2.5}$ and lower neurodevelopment in children ranging in age from 6 months to 6 years.[12 35 40–42] In agreement with our findings, a multicentre cohort study from six European countries investigated the effects of prenatal exposure to multiple air pollutants including $PM_{2.5}$, $PM_{10}$, coarse particles, $NO_2$ and nitrogen oxides (NOx) among 9482 children between 1 and 6 years; the authors found non-significant positive associations between prenatal $PM_{2.5}$ exposure and normal neurodevelopment (β: 1.64, 95% CI: −3.47 to 0.18; per 5 μg/m³ increase in $PM_{2.5}$).[34] Similarly, another study examining the effects of multiple pollutant exposures on early childhood cognition at 40 days of age in a highly exposed area of Spain also found $PM_{10}$, $PM_{coarse}$, $PM_{2.5absorbance}$, $NO_2$, $NO_x$ and ozone were linked to lower motor function in children, except for $PM_{2.5}$.[43] The inconsistent findings could be because of heterogeneity between studies in terms of exposure (eg,

exposure assessment methods used, $PM_{2.5}$ exposure levels or composition of $PM_{2.5}$).

The prevalence of MDD and PDD in our study is higher than in other studies that also used the CBSID to report developmental delay rates, which were at 17%,[44] 15.78%[45] and 13.68%.[46] This may be attributed to the younger age of infants in our study, which were assessed at around 12 months, compared with most studies assessed at around 24 months. A Chinese study and a South Korean study also found lower scores on the MDI and PDI in 1-year-old children.[47 48] Aside from the conflicting findings regarding prenatal $PM_{2.5}$ exposure and neurodevelopmental outcomes, results regarding the most potential sensitive time windows before and during pregnancy are also inconclusive. Some studies suggested that early-to-mid pregnancy might be a potential sensitive period,[21 22] while other studies found stronger associations for middle-to-late pregnancy, thus results are equivocal.[20 24 25]

The potential biological mechanisms by which air pollution could affect neurodevelopment are not yet clearly understood. There is evidence suggesting that exposure to prenatal $PM_{2.5}$ could potentially induce maternal immune activation during pregnancy.[49] Higher levels of cytokines or reactive oxygen species may potentially interfere with fetal neurodevelopment through three mechanisms: crossing the placental barrier into the fetal body, inducing fetal immune dysregulation and contributing to inadequate placental perfusion that affects nutritional processes and oxygenation of maternal blood.[50] More research is needed to investigate the trimester effects of air pollution on neurodevelopment and provide a better understanding of the underlying biological mechanisms. Our study is the first to consider an exposure window 90 days prior to conception for $NO_2$. A novel observation is that the effects of $NO_2$ or $PM_{2.5}$ air pollution on child cognition can be seen at least 90 days prior to conception, representing a potentially vulnerable periods in relation to air pollution on neurodevelopment. Similar results were found in a previous study that recruited 1329 mother–child pairs in Wuhan, China.[12] This study reported a higher level of $PM_{2.5}$ during preconception (median: 76.1 µg/m³) and in the first trimester (median: 82.3 µg/m³). This study found for each doubling of $PM_{2.5}$ exposure during preconception, children's PDI scores was reduced by 6.15 (95% CI: −8.84 to −3.46) points. A potential explanation is that preconception air pollution exposures induce genetic and epigenetic alterations in sperm, that increase the risk of adverse health outcomes in offspring.[51 52] To date, all studies examined the effect of maternal preconception exposure while omitting paternal exposures.[17] Future studies should consider the effect of preconception paternal exposure in relation to childhood health outcomes.

This study has several strengths. We developed an LUR model to capture spatial and temporal variations of air pollution at individual level to reduce exposure misclassification if using monitoring stations. This is a novel study to investigate both pre-conception and prenatal $PM_{2.5}$

and $NO_2$ exposure with neurodevelopment outcomes among young infants, in the context of a relatively high air pollution urban environment. The exposure levels in our study were similar to those in comparable urban areas in Chinese cities. A study in Shanghai, China reported an average $NO_2$ exposure during pregnancy from 2014 to 2015, predicted by the LUR model, of 48.23 µg/m³ (mean $PM_{2.5}$ in our study: 50.52 µg/m³).[53] Similarly, a study in Tianjin found the annual average $PM_{2.5}$ exposure to be 62 µg/m³ in 2017 (mean $NO_2$ in our study: 57.48 µg/m³).[54] Wu et al developed an LUR model for $PM_{2.5}$ in the main urban area of Chongqing.[55] This model predicted an annual average $PM_{2.5}$ concentration of 40.6 µg/m³,[55] whereas our prediction is higher at 55.9 µg/m³.[19] This difference can be attributed to the temporal variations. Wu et al used monitoring data from 2013, while we used data from 2015. It could be considered that our GAM model, with its temporal component, could explain temporal variations and is more suitable for pregnancy-specific exposure estimates.

A major limitation of this study was that our sample size was relatively small, limiting the statistical power to assess several outcomes, although the higher exposures in Chongqing than in some other studies may increase the probability of detecting effects. In terms of limitations, due to a lack of information on participant time-activity patterns, exposure estimates in this study refer only to ambient concentrations at home addresses and no other activity spaces (eg, indoor, workplace, commuting) were considered. We may have thus underestimated total air pollution exposure. Second, we defined exposure windows for clinically-defined trimesters; sensitive periods may be shorter or longer than 3 months, or they may exist in the overlap of multiple trimesters. However, we were unable to investigate the sensitive time windows using established methods such as distributed lag non-linear models due to the lack of highly time-resolved air pollution estimates. Third, the performance of the $NO_2$ spatiotemporal model was low (COR-$R^2$: 0.39), which may introduce exposure misclassification and therefore bias in the coefficients. It may lead to underestimation of the association if the $NO_2$ spatiotemporal model inadequately represents the true variability in $NO_2$ levels. Or conversely, it could overestimate the association between $NO_2$ exposure and the outcome if the model fails to account for certain factors or inaccurately estimates $NO_2$ levels. Finally, we were unable to include some other air pollutants such as polycyclic aromatic hydrocarbons, black carbon and $O_3$, which have been found particularly harmful to neurodevelopment in children.[56] Although we have accounted for most of the important confounders in this study, unfortunately, we did not collect information on the feeding patterns of infants. This may undermine the validity and reliability of our findings.

## CONCLUSION

This study provides evidence for an association between $NO_2$ exposure prior to and during pregnancy with birth and neurodevelopmental outcomes in a birth cohort in Chongqing, China. Exposure to $NO_2$ and $PM_{2.5}$ exposure before pregnancy was associated with a lower psychomotor development score. Increased $NO_2$ exposure was linked to a risk of psychomotor development delay during various pregnancy trimesters.

**Author affiliations**

[1]Centre for Environmental Health and Sustainability, University of Leicester, Leicester, UK

[2]Department of Public Health and Management, Zunyi Medical and Pharmaceutical College, Zunyi, China

[3]Stomatological Hospital of Chongqing Medical University, Chongqing, China

[4]Department of Health Sciences, University of Leicester, Leicester, UK

[5]University of Kansas, Lawrence, Kansas, USA

[6]University of Leicester, Leicester, UK

[7]University of Auckland Liggins Institute, Auckland, New Zealand

[8]Canada - China -New Zealand Joint Laboratory of Maternal and Fetal Medicine, Chongqing, China

[9]School of Public Health, Chongqing Medical University, Chongqing, China

[10]St George's University of London, London, UK

[11]Department of Obstetrics and Gynaecology, The First Affiliated Hospital of Chongqing Medical University, Chongqing, China

[12]College of Medicine, University of Leicester, Leicester, UK

**Acknowledgements** The authors would like to acknowledge the clinical research staff who recruited subjects and facilitated sample collection, the women and their families who participated in the Complex Lipids in Mothers and Babies (CLIMB) study, and Jamie de Seymour, the leading author for diet pattern analysis in the CLIMB cohort, for her help and advice during the analysis process.

**Contributors** YX, T-LH, HZ and PB conceived and designed research. TZ, YX and HZ recruited the patients and collected the samples. TK, AH and JG constructed the air pollution model. YC analysed, interpreted the data and prepared the figures. YC and TK were major contributors in writing the manuscript text. YC, JC, T-LH, YX, AH and PB substantively revised the manuscript. All authors read and approved the final manuscript. YX is the guarantor of the manuscript.

**Funding** This work was supported by the National Natural Science Foundation of China (No. 81971406, 82271715), The 111 Project (Yuwaizhuan (2016)32), Chongqing Science & Technology Bureau (CSTB2022NSCQ-MSX1680), Youth Innovation Team Development Support Program of Chongqing Medical University (W0083) and Smart Medicine Research Project of Chongqing Medical University (No. ZHYX202103), Zunyi science and technology plan project (Zunshikehe HZ (2022)153). The research was supported by National Institute for Health Research (NIHR) Health Protection Research Unit in Environmental Exposures and Health, a partnership between UK Health Security Agency, the Health and Safety Executive and the University of Leicester and by the NIHR Leicester Biomedical Research Centre (BRC). The views expressed are those of the author(s) and not necessarily those of the NIHR, the Department of Health and Social Care or UK Health Security Agency.

**Map disclaimer** The inclusion of any map (including the depiction of any boundaries therein), or of any geographical or locational reference, does not imply the expression of any opinion whatsoever on the part of BMJ concerning the legal status of any country, territory, jurisdiction or area or of its authorities. Any such expression remains solely that of the relevant source and is not endorsed by BMJ. Maps are provided without any warranty of any kind, either express or implied.

**Competing interests** None declared.

**Patient and public involvement** Patients and/or the public were not involved in the design, or conduct, or reporting, or dissemination plans of this research.

**Patient consent for publication** Consent obtained directly from patient(s).

**Ethics approval** Ethical approval for this study was granted by the Ethics Committee of Chongqing Medical University (#2014034). The participants provided their written informed consent to participate in this study. Written informed consent was obtained from the individual(s) for the publication of any potentially identifiable images or data included in this article.

**Provenance and peer review** Not commissioned; externally peer reviewed.

**Data availability statement** Data may be obtained from a third party and are not publicly available. Not applicable.

**ORCID iDs**

Yingxin Chen http://orcid.org/0000-0002-1567-2606

Yinyin Xia http://orcid.org/0000-0001-6536-1868

Anna Hansell http://orcid.org/0000-0001-9904-7447

Hua Zhang http://orcid.org/0000-0003-3501-8385

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
