## [Reviewer comments · BMJ Open]

ARTICLE DETAILS

TITLE (PROVISIONAL)	Associations of air pollution exposures in preconception and pregnancy with birth outcomes and infant neurocognitive development: analysis of the Complex Lipids in Mothers and Babies (CLIMB) prospective cohort in Chongqing, China
AUTHORS	Chen, Yingxin; Kuang, Tao; Zhang, Ting; Cai, Samuel; Colombo, John; Harper, Alex; Han, Ting-Li; Xia, Yinyin; Gulliver, John; Hansell, Anna; Zhang, Hua; Baker, Philip

VERSION 1 – REVIEW

REVIEWER	Zhang, Zilong
REVIEW RETURNED	26-Dec-2023

GENERAL COMMENTS	The authors examined the associations between maternal exposure to air pollution before conception and during pregnancy and birth outcomes and infant neurodevelopment in a birth cohort in Chongqing. Overall, this is an important study with some interesting findings. I have several suggestions. 1. The title seems a bit inappropriate as the authors actually examined the air pollution exposure at different periods including preconception and pregnancy stages, which are more than “early pregnancy”.2. In Introduction, the authors should provide a better rationale for focusing on the selected two pollutants, especially for NO₂. The authors stated that NO₂ was less investigated, but this does not necessarily mean it is an essential pollutant requiring extensive investigations.3. I would not consider “lack of evidence from developing countries” as a major research gap. The authors listed quite a lot studies from these regions, particularly China.4. The authors claimed that identification of sensitive time windows was critical but did not tackle this issue with a finer approach such as DLNM. Well, the authors argued that they were unable to do so because of the lack of highly time-resolved air pollution estimates. This is not convincing, as the LUR provided daily estimates according to the relevant introduction in Methods section.5. For the neurodevelopment, did the authors consider binary outcomes (e.g., using a cut-off to define neurodevelopmental delay) in addition to the scores?
---

	6. Some important potential confounders were missing, such as dietary supplements during pregnancy especially folic acid, and feeding pattern for infants. 7. I am very confused that the authors used different levels of statistical significance in different analyses but provided no justification. Would suggest to use the 0.05 only and keep consistent. 8. It is a bit strange to observe negative correlations between some exposures. Please comment. 9. Overall, no significant associations were found between PM2.5/NO2 exposure and birth outcomes. As the two-pollutant models were subject to collinearity, findings from single-pollutant models were more reliable and should be considered as main findings. Consequently, the relevant conclusions should be modified. 10. The mediation analysis made no sense to me as no significant associations were identified between exposures and the mediators. 11. The authors claimed the positive association between PM2.5 and PDI was unexpected, but did not provide sufficient discussion about the possible reasons, especially those from biological perspectives. 12. The language could be improved as numerous grammar and wording errors were present. Also, the format of tables could be improved to present the results more clearly.
--	---

REVIEWER	Pratiti , Rebecca McLaren Health Care Corp, Internal Medicine
REVIEW RETURNED	18-Jan-2024

GENERAL COMMENTS	The manuscript is well written. Statistics seems appropriate. I could not comment about the statistics involving mediation effect. The number of tables may be decreased. It may be beneficial to comment in discussion section about the new generalized additive model discussed in this study. How the exposure of PM2.5 or NO in this study varied from levels estimated in other studies from same city or similar urban areas to evaluate if the new GAM model overestimated or underestimated the exposure.
--

VERSION 1 – AUTHOR RESPONSE

Reviewer: 1
Zilong Zhang

Comments to the Author:

The authors examined the associations between maternal exposure to air pollution before conception

and during pregnancy and birth outcomes and infant neurodevelopment in a birth cohort in Chongqing. Overall, this is an important study with some interesting findings. I have several suggestions.

1. The title seems a bit inappropriate as the authors actually examined the air pollution exposure at different periods including preconception and pregnancy stages, which are more than “early pregnancy”.

We acknowledge your point regarding the appropriateness of the title considering the broader scope of air pollution exposure periods examined in our study, including preconception and pregnancy stages. The revised title now reads: "Associations of air pollution exposures in preconception and pregnancy with birth outcomes and infant neurocognitive development: analysis of the Complex Lipids in Mothers and Babies (CLIMB) prospective cohort in Chongqing, China".

2. In Introduction, the authors should provide a better rationale for focusing on the selected two pollutants, especially for NO₂. The authors stated that NO₂ was less investigated, but this does not necessarily mean it is an essential pollutant requiring extensive investigations.

We have carefully considered your suggestion and incorporated additional information to provide the rationale for focusing on the selected pollutants, especially NO₂, in the ‘*Introduction*’ section.

" Exposure to NO₂ during pregnancy may be linked to compromised neural development in children, particularly affecting fine psychomotor skills (16). Studying PM_{2.5} along with NO₂ may allow us to explore how multiple pollutants affect birth outcomes and infant neurocognitive development independently and jointly. Moreover, both PM_{2.5} and NO₂ are regulated traffic-related air pollutants in many countries. Understanding their impacts on birth and infant neurocognitive development can provide valuable insights for policymakers and public health authorities to develop effective air quality regulations and interventions."

We believe this addition enhances the rationale for examining both PM_{2.5} and NO₂ in our study, emphasizing their potential individual and joint effects on birth outcomes and infant neurocognitive development.

3. I would not consider “lack of evidence from developing countries” as a major research gap. The authors listed quite a lot studies from these regions, particularly China.

We have revised the manuscript accordingly by removing the statement regarding the "lack of evidence from developing countries" as a major research gap. We appreciate your input in refining the clarity and focus of our study.

4. The authors claimed that identification of sensitive time windows was critical but did not tackle this

issue with a finer approach such as DLNM. Well, the authors argued that they were unable to do so because of the lack of highly time-resolved air pollution estimates. This is not convincing, as the LUR provided daily estimates according to the relevant introduction in Methods section.

Thank you for your feedback regarding the identification of sensitive time windows and the use of a finer approach such as Distributed Lag Non-linear Models (DLNM) in our study. We agree this would have been a useful analysis. While the original LUR models provided daily air pollution estimates, unfortunately, the analyst for the LUR models, has graduated and left academia and his supervisor has since moved institutions. Unfortunately, following the moves, the original model outputs have been deleted from university systems so we only have access to the trimester averages as used in this paper.

5. For the neurodevelopment, did the authors consider binary outcomes (e.g., using a cut-off to define neurodevelopmental delay) in addition to the scores?

We have incorporated an analysis using a cut-off score of 85 to define neurodevelopmental delay, in addition to the scores previously used. In the '*Neurodevelopment outcomes*' section, we added the following statement: " In addition to the continuous scores, we define mental developmental delay (MDD) and psychomotor developmental delay (PDD) if the score is less than 85 (32) " In the '*Statistical analysis*' section, we added the following statement: "We also conducted multivariable logistic regression analysis for binary neurodevelopment outcomes (i.e., MDD and PDD)." And we have revised our results and discussion accordingly.

6. Some important potential confounders were missing, such as dietary supplements during pregnancy especially folic acid, and feeding pattern for infants.

Thank you for your insightful comments regarding potential confounders in our study. Regarding dietary supplements during pregnancy, particularly folic acid intake. In our cohort, all pregnant women routinely take folic acid. Folic acid is provided free of charge by community hospitals. Women who are planning pregnancy typically obtain folic acid from these hospitals. Additionally, during their first visit, doctors inquire about folic acid intake, and if women are not taking it, they are advised to do so. We have added a statement in '*Method*' section: "We did not adjust dietary supplements during pregnancy because all pregnant women routinely take folic acid in this cohort." As for feeding patterns for infants, unfortunately, we did not collect this information in our study, which we have mentioned this in '*Discussion*' section as a study limitation: "Although we have accounted for most of the important confounders in this study, unfortunately, we did not collect information on the feeding patterns of infants. This may undermine the validity and reliability of our findings."

7. I am very confused that the authors used different levels of statistical significance in different analyses but provided no justification. Would suggest to use the 0.05 only and keep consistent.

In the updated version, we have revised our approach to maintain a consistent significance level of 0.05 across all analyses.

8. It is a bit strange to observe negative correlations between some exposures. Please comment.

We have revised the manuscript accordingly by adding the following statement in the '*Discussion*' section: "We also observed negative correlations between certain exposures, indicating the need to consider potential collinearity in our two-pollutant models. In Chongqing, a major industrial city in southwest China, air pollution may come from industrial and traffic emissions, construction activities, and dust, and negative correlations may occur if different sources contribute disproportionately to each pollutant. Their correlations may also be affected by seasonal changes and variations in weather patterns. Future research should also explore the impact of different sources of air pollution on children's cognitive health."

9. Overall, no significant associations were found between PM_{2.5}/NO₂ exposure and birth outcomes. As the two-pollutant models were subject to collinearity, findings from single-pollutant models were more reliable and should be considered as main findings. Consequently, the relevant conclusions should be modified.

Thank you for highlighting this. We have made the following revisions:

In the '*Abstract*' section: "Conclusions: Increased exposure to NO₂ during pregnancy were associated with increased risk of SGA and psychomotor development delay, while increased exposure to both PM_{2.5} and NO₂ pre-conception were associated with adverse psychomotor development outcomes at 12 months of age.."

In the '*Conclusion*' at the end of the main text: "This study provides evidence for an association between NO₂ exposure prior to- and during pregnancy with birth and neurodevelopmental outcomes in a birth cohort in Chongqing, China. Exposure to NO₂ and PM_{2.5} exposure before pregnancy was associated with a lower psychomotor development score. Increased NO₂ exposure was linked to a risk of psychomotor development delay during various pregnancy trimesters."

10. The mediation analysis made no sense to me as no significant associations were identified between exposures and the mediators.

Upon careful consideration of your comments, we have decided to remove the mediation analysis from our study. We agreed that without significant associations between exposures and the mediators, conducting a mediation analysis may not be meaningful. We have revised our manuscript accordingly.

11. The authors claimed the positive association between PM_{2.5} and PDI was unexpected, but did not provide sufficient discussion about the possible reasons, especially those from biological perspectives.

We have included additional discussion in the main text regarding the unexpected positive association between PM_{2.5} and PDI. In the revised '*Discussion*' section, we have added: " Contrary to expectations, we found significant positive associations between prenatal exposure to PM_{2.5} air pollution in the second trimester and PDI. However, no association was observed between PM_{2.5} exposures in the second trimester and the risk of PDD. Given the existing literature and the conflicted observation here, we believe that this is likely to be spurious/sample specific. Some plausible explanations include the uneven distribution of PDI scores, the potentially inappropriate selection of the cut-off value of 85 (which may not effectively discriminate between groups), or the possibility that the observed outcome occurred by chance. " We hope this added discussion addresses the reviewer's concerns and provides further insight into the unexpected findings.

12. The language could be improved as numerous grammar and wording errors were present. Also, the format of tables could be improved to present the results more clearly.

We sincerely apologize for any grammatical or wording errors that may have detracted from the clarity of our work. We thoroughly reviewed the manuscript and conducted multiple rounds of proofreading by native speakers to address any remaining language issues and to ensure that the text is clear, concise, and grammatically correct. Additionally, we enhanced the format of tables to present the results more effectively, and we ensured that all tables are integrated within the text shortly after they are first cited.

Your input is greatly appreciated, and we are committed to making the necessary revisions to enhance the quality of our manuscript.

Reviewer: 2

Dr. Rebecca Pratiti , McLaren Health Care Corp

Comments to the Author:

The manuscript is well written. Statistics seems appropriate. I could not comment about the statistics involving mediation effect.

We have made the decision to remove the mediation analysis from our study and have adjusted our manuscript accordingly. We understand that without significant associations between exposures and the mediators, conducting a mediation analysis may not be meaningful.

The number of tables may be decreased.

We have removed some unnecessary tables upon request. In the previous version, there were 3 tables in the main text and 11 in the supplement, while in this version there are 4 in the main text and 7 in the supplement.

It may be beneficial to comment in discussion section about the new generalized additive model discussed in this study. How the exposure of PM_{2.5} or NO in this study varied from levels estimated in other studies from same city or similar urban areas to evaluate if the new GAM model overestimated or underestimated the exposure.

Thank you for your suggestion. In the ‘*discussion*’ section, we have provided insights regarding the new generalized additive model (GAM) employed in our study. We compared the exposure levels of PM_{2.5} and NO₂ in our study with those estimated in other studies conducted in Shanghai and Tianjin, two cities in China and another LUR model developed in Chongqing. We added the following statement in the ‘*discussion*’ section: “The exposure levels in our study were similar as those in comparable urban areas in Chinese cities. A study in Shanghai, China reported an average NO₂ exposure during pregnancy from 2014 to 2015, predicted by the LUR model, of 48.23 µg/m³ (Mean PM_{2.5} in our study: 50.52 µg/m³) (47). Similarly, a study in Tianjin found the annual average PM_{2.5} exposure to be 62 µg/m³ in 2017 (Mean NO₂ in our study: 57.48 µg/m³) (48). Wu et al. developed a LUR model for PM_{2.5} in the main urban area of Chongqing (49). This model predicted an annual average PM_{2.5} concentration of 40.6 µg/m³ (49), whereas our prediction is higher at 55.9 µg/m³ (19). This difference can be attributed to the temporal variations. Wu et al. used monitoring data from 2013, while we utilized data from 2015. It could be considered that our GAM model, with its temporal component, could explain temporal variations and is more suitable for pregnancy-specific exposure estimates.”

We would like to thank all editors and reviewers for your comments and suggestions. I believe that this work has been revised accordingly and improved after the major revision.

VERSION 2 – REVIEW

REVIEWER	Zhang, Zilong
REVIEW RETURNED	27-Mar-2024
GENERAL COMMENTS	The authors have responded to my comments well.
REVIEWER	Pratiti, Rebecca McLaren Health Care Corp, Internal Medicine
REVIEW RETURNED	12-Apr-2024
GENERAL COMMENTS	The authors have addressed most the reviewer comments. The manuscript is clearer. Some comments to improve the clarity of the manuscript. 1. Page 8, line 12-15: Please clarify if the women were excluded based on self-stated history or per physician evaluation.

	2. Page 9, line 18-23: It would be incorrect to mention NO₂ spatiotemporal model is moderate since COR-R² is less than 0.4. And hence is a low correlation model. Though it is mentioned in the limitation section, it needs to be elaborated how it could have affected the outcome estimate. Under or overestimate the association. 3. Page 12, line 16-21: The MDD and PDD prevalence in this is 27% and 42% that is significantly higher than normal and needs to be discussed in the discussion section. Possible also compare with other studies. The prevalence of PDD could be explained for by the PM_{2.5} and NO₂, though the prevalence of MDD could not be detected and possible explanation should be provided for the high MDD. If, possible try to remove season of birth from model and evaluate the association. 4. Page 12, line 35-41: There is a high variability of PM_{2.5} in this study. Chongqing is an industrial city with multiple old industries including manufacturing, hence it may be good to add a section on study setting to the Methods section. Elaborating the industries present in Chongqing, those specifically with higher NO₂ and PM_{2.5} emissions, average automobile, population density and possibly other sources for NO₂ emission in this region. Also, the prevalence of biomass cookstoves in this area. Changes in the composition and sources of PM 2.5 causes modification to its resulting toxicity, mutagenicity and oxidative potential. 5. Table 1: 80.5% of the study women were working full time. If possibly would be good to add in limitation, that it was not measured how many were in industrial job. Though there is already a comment in limitation not measuring other sources of air pollution. 6. Page 19: The association were adjusted for seasons of birth. It would be good to mention somewhere in introduction or methods about why it was accounted for and how it affects outcome estimate. 7. Page 20, line 88: I guess it should be positive correlation with negative impact. 8. Page 35: in the last box, number of new-borns included is not mentioned. 9. Page 49, eTable7: The yes/no of the Primiparity is not adding up within each group of included and excluded.
--	--

VERSION 2 – AUTHOR RESPONSE

Reviewer: 1

Zilong Zhang

Comments to the Author:

The authors have responded to my comments well.

Thank you for your positive feedback regarding our responses to your previous comments. Your feedback has been invaluable in improving the quality of our work. We are pleased to hear that you found our revisions satisfactory.

Reviewer: 2

Dr. Rebecca Pratiti , McLaren Health Care Corp

Comments to the Author:

The authors have addressed most the reviewer comments. The manuscript is clearer. Some comments to improve the clarity of the manuscript.

1. Page 8, line 12-15: Please clarify if the women were excluded based on self-stated history or per physician evaluation.

Thank you for your comment. The women were excluded based on self-stated history, and we have revised the main text accordingly. It now states, "Women were excluded if they had a self-stated history of premature delivery before 32 weeks of gestation, maternal milk allergy or aversion, or severe lactose intolerance."

2. Page 9, line 18-23: It would be incorrect to mention NO₂ spatiotemporal model is moderate since COR-R² is less than 0.4. And hence is a low correlation model. Though it is mentioned in the limitation section, it needs to be elaborated how it could have affected the outcome estimate. Under or overestimate the association.

We acknowledge your point that a correlation coefficient (COR-R²) of less than 0.4 signifies a low correlation model rather than a moderate one. Accordingly, we have revised the '*Exposure assessment*' section to reflect this." The performance of the PM_{2.5} spatiotemporal models was good (Correlation (COR)-R²: 0.72) and the NO₂ spatiotemporal model was low (COR-R²: 0.39) when providing concentration estimates in absolute terms."

We have also included the following statement in the '*limitations*' section to elaborate on how it could have affected the outcome estimate in both ways: "It may lead to underestimation of the association if the NO₂ spatiotemporal model inadequately represents the true variability in NO₂ levels. Or conversely, it could overestimate the association between NO₂ exposure and the outcome if the model fails to account for certain factors or inaccurately estimates NO₂ levels."

3. Page 12, line 16-21: The MDD and PDD prevalence in this is 27% and 42% that is significantly higher than normal and needs to be discussed in the discussion section. Possible also compare with other studies. The prevalence of PDD could be explained for by the PM_{2.5} and NO₂, though the prevalence of MDD could not be detected and possible explanation should be provided for the high MDD. If, possible try to remove season of birth from model and evaluate the association.

Thank you for bringing this to our attention. Upon further review, we acknowledge that the prevalence of MDD and PDD in our study is higher than expected. However, it's important to note that there are limited studies utilizing a cut-off value to define neurodevelopmental delay for CBSID, with most studies using MDI and PDI as continuous variables. Among these studies, the infants were typically around two years old, which may explain the higher prevalence observed in our study due to the younger age of our participants.

In response to your suggestion, we have added a few clarifications into discussion: "The prevalence of MDD and PDD in our study is higher than in other studies that also used the CBSID to report developmental delay rates, which were at 17% (46), 15.78% (47), and 13.68% (48). This may be

attributed to the younger age of infants in our study, which were assessed at around 12 months, compared to most studies assessed at around 24 months. A Chinese study and a South Korean study also found lower scores on the MDI and PDI in 1-year-old children (49, 50).”

4. Page 12, line 35-41: There is a high variability of PM_{2.5} in this study. Chongqing is an industrial city with multiple old industries including manufacturing, hence it may be good to add a section on study setting to the Methods section. Elaborating the industries present in Chongqing, those specifically with higher NO₂ and PM_{2.5} emissions, average automobile, population density and possibly other sources for NO₂ emission in this region. Also, the prevalence of biomass cookstoves in this area. Changes in the composition and sources of PM 2.5 causes modification to its resulting toxicity, mutagenicity and oxidative potential.

Thank you for your insightful comment and the suggestion. We have added "*Study setting*" section in the Methods: “The study area focused on the urban center of the Chinese municipality of Chongqing (Figure 2). The terrain of Chongqing is predominantly hilly and mountainous, with the core area located in a synclinal valley at the confluence of the Yangtze River and the Jialing River (28). The urban core of Chongqing, our study area, has a population of approximately 6.52 million people, a land area of 5,472 square kilometers, and 4.62 million vehicles (29). It shows a higher population density of approximately 1,191 people per square kilometer and a lower number of motor vehicles of 0.71 per capita. The urban core of Chongqing used to have multiple old industries with higher NO₂ and PM_{2.5} emissions, including the Chongqing Iron and Steel Company in Dadukou District and the Chongqing Thermal Power Plant in Jiulongpo District, both of which have been relocated to rural areas in Chongqing. The main sources of pollution in the area now include traffic-related emissions, construction activities, and anthropogenic sources such as outdoor grilling and emissions from food establishments (30). The coverage rate of urban population with access to gas in Chongqing was 95.34% (29), suggesting a low reliance on biomass cookstoves in urban areas.”

5. Table 1: 80.5% of the study women were working full time. If possibly would be good to add in limitation, that it was not measured how many were in industrial job. Though there is already a comment in limitation not measuring other sources of air pollution.

Thank you for your comment. None of the women included in our study were employed in industrial jobs.

6. Page 19: The association were adjusted for seasons of birth. It would be good to mention somewhere in introduction or methods about why it was accounted for and how it affects outcome estimate.

Thank you for your valuable suggestion. We have incorporated an explanation in the ‘Covariates’ section regarding the adjustment for seasons of birth, detailing the rationale behind its consideration: “Season of birth was taken into consideration because air pollution and related environmental factors, such as temperature and humidity, may vary across different seasons (i.e., air pollution levels tend to be higher during winter). Some studies suggest that the season of birth may indirectly influence cognitive function through factors such as seasonal differences in food availability affecting maternal nutrition during pregnancy, sunlight exposure impacting maternal vitamin D levels, and children’s early-life indoor and outdoor activities.”

7. Page 20, line 88: I guess it should be positive correlation with negative impact.

Thank you for pointing this out. This sentence has now been revised as follows: “Many studies from other geographic areas, including Europe (36-38), the United States (22, 26), and Asia (23, 39-41), have found significant associations between prenatal air pollution exposure and a variety of adverse neurodevelopmental outcomes.”

8. Page 35: in the last box, number of new-borns included is not mentioned.

Thank you for pointing this out. We noticed that some content in this figure was missing due to format conversion. It has now been modified and all numbers have been double-checked.

9. Page 49, eTable7: The yes/no of the Primiparity is not adding up within each group of included and excluded.

Thank you for pointing this out. After checking, we found that we unintentionally copied the wrong number into the wrong square. We apologize for any confusion it may have caused, and we have now corrected all numbers.

VERSION 3 – REVIEW

REVIEWER	Pratiti, Rebecca McLaren Health Care Corp, Internal Medicine
REVIEW RETURNED	06-Jun-2024
GENERAL COMMENTS	The authors have addressed the reviewer comments.